# A Cationic Zn-Phthalocyanine Turns Alzheimer’s Amyloid β Aggregates into Non-Toxic Oligomers and Inhibits Neurotoxicity in Culture

**DOI:** 10.3390/ijms25168931

**Published:** 2024-08-16

**Authors:** Abdullah Md. Sheikh, Shatera Tabassum, Shozo Yano, Fatema Binte Abdullah, Ruochen Wang, Takahisa Ikeue, Atsushi Nagai

**Affiliations:** 1Department of Laboratory Medicine, Shimane University Faculty of Medicine, 89-1 Enya Cho, Izumo 693-8501, Japan; tabassum@med.shimane-u.ac.jp (S.T.); syano@med.shimane-u.ac.jp (S.Y.); anagai@med.shimane-u.ac.jp (A.N.); 2Department of Neurology, Shimane University Faculty of Medicine, 89-1 Enya Cho, Izumo 693-8501, Japan; m239401@med.shimane-u.ac.jp (F.B.A.); m239606@med.shimane-u.ac.jp (R.W.); 3Department of Chemistry, Graduate School of Science and Engineering, Shimane University, 1060 Nishikawatsu, Matsue 690-8504, Japan; ikeue@riko.shimane-u.ac.jp

**Keywords:** amyloid β, fibril formation, amyloid β binding, oligomers, neurodegeneration, Alzheimer’s disease

## Abstract

Amyloid β peptide (Aβ) aggregation and deposition are considered the main causes of Alzheimer’s disease. In a previous study, we demonstrated that anionic Zn-phthalocyanine (ZnPc) can interact with the Aβ peptide and inhibit the fibril-formation process. However, due to the inability of anionic ZnPc to cross the intact blood–brain barrier, we decided to explore the interaction of cationic methylated Zn-phthalocyanine (cZnPc) with the peptide. Using a ThT fluorescence assay, we observed that cZnPc dose-dependently and time-dependently inhibited Aβ_1-42_ fibril levels under in vitro fibril-formation conditions. Electron microscopy revealed that it caused Aβ_1-42_ peptides to form small aggregates. Western blotting and dot immunoblot oligomer experiments demonstrated that cZnPc increased rather than decreased the levels of oligomers from the very early stages of incubation. A binding assay confirmed that cZnPc could bind with the peptide. Docking simulations indicated that the oligomer species of Aβ_1-42_ had a higher ability to interact with cZnPc. ANS fluorescence assay results indicated that cZnPc did not affect the hydrophobicity of the peptide. However, cZnPc significantly increased intrinsic tyrosine fluorescence of the peptide after 8 h of incubation in fibril-formation conditions. Importantly, cell culture experiments demonstrated that cZnPc did not exhibit any toxicity up to a concentration of 10 µM. Instead, it protected a neuronal cell line from Aβ_1-42_-induced toxicity. Thus, our results suggest that cZnPc can affect the aggregation process of Aβ_1-42_, rendering it non-toxic, which could be crucial for the therapy of Alzheimer’s disease.

## 1. Introduction

Alzheimer’s disease (AD) is a chronic neurodegenerative disorder in the elderly population that is clinically characterized by a gradual decrease in cognitive functions, including memory [1]. It initially presents as subtle changes in higher cortical functions, including behavior, mood, and decision-making processes, followed by noticeable memory loss, altered speech, and aphasia [1]. Pathologically, the disease first affects the hippocampus, entorhinal cortex, and amygdala, and later extends to other brain regions such as the frontal, parietal, and temporal lobes, showing varying degrees of cortical atrophy [2]. Microscopically, the diagnostic criteria for AD include the presence of neuritic plaques and intraneuronal neurofibrillary tangles, along with some degree of neuroinflammation in the lesion area [2]. Neuritic plaque areas exhibit excessive accumulation of the amyloid β (Aβ) peptide, along with dystrophic neurites and activated microglia [2].

Several key proteins play a pivotal role in modulating the pathology of AD (reviewed in Figure 1). As mentioned before, extracellular deposition of Aβ peptides is considered a hallmark of AD. Aβ is a peptide consisting of 39–43 amino acids, generated from the amyloid precursor protein (APP) through the activity of β- and γ-secretase enzymes [2,3]. Numerous preclinical and clinical studies have established a link between Aβ and AD pathology. Genetic mutations in APP and its processing enzymes, which increase Aβ production, are the primary cause of familial AD and AD-like features in Down syndrome [4,5]. Transgenic mice that overexpress mutated APP and Aβ-producing enzymes demonstrate many AD-like features, including Aβ deposition in the hippocampal and cortical areas, neurodegeneration, neuroinflammation, phospholipid changes, and memory loss [6,7]. In sporadic AD, the protein apolipoprotein E (ApoE), particularly its ε4 allele, is genetically associated with the disease [8,9]. Later it was found that the ε4 isoform of ApoE impedes the clearance of Aβ [10]. Tau protein is another critical player in AD pathology [11]. It undergoes abnormal hyperphosphorylation, resulting in the formation of neurofibrillary tangles within neurons [11] These tangles disrupt the microtubule network, impairing axonal transport and contributing to neuronal dysfunction and degeneration [11,12,13]. Importantly, it has been shown that Aβ can increase the phosphorylation of tau [14]. Additionally, dysregulation of the protein quality control system, including chaperones, proteasomes, and autophagy-related proteins, has been shown to be important in AD pathology by affecting Aβ clearance and tau aggregation into neurofibrillary tangles [15]. Then, the accumulation of Aβ induces a vicious cycle where Aβ affects the protein quality control system, resulting in further increases in Aβ accumulation, tau phosphorylation, and aggregation. This leads to neurofibrillary tangle formation, neuroinflammation, and neurodegeneration, culminating in the pathological and clinical manifestations of AD. Hence, these findings suggest that Aβ occupies a vital position in AD pathology.

In AD brains, Aβ is found to aggregate in the lesion areas [16,17]. Low levels of Aβ peptides are continuously produced and cleared from the brain, which suggests that its aggregation process could be crucial for deposition. Since the deposited peptides are in aggregated form, several in vitro studies have explored the aggregation process of the peptide and its consequences, showing that the concentration of the peptide, time, temperature, pH, and presence of other macromolecules and proteins, including apoE4, can influence the process [18,19]. Various aggregated peptide species, including oligomers, protofibrils, or mature fibrils, are observed during aggregation kinetics. In vitro cell culture studies have demonstrated that larger-size soluble oligomers are highly neurotoxic and neuroinflammatory, features commonly seen in AD brains [20]. These findings suggest that Aβ aggregation is an important process in AD pathology and could be a suitable target for therapy.

For AD therapy, several studies have targeted aspects of Aβ pathology, including production, clearance, and aggregation processes [21,22]. The potentials of Aβ-producing enzyme inhibitors and antibody-mediated clearance have been explored. However, β-secretase inhibitors have shown many adverse effects [23], and antibody-mediated AD therapy has demonstrated several unwanted immunological effects [24]. Hence, amyloid aggregation inhibitors could be a safer and more promising option. Previously, we investigated a hydrophilic anionic carboxylate Zn-phthalocyanine (ZnPc) for its ability to interact with and inhibit Aβ aggregation [25]. We found that it effectively inhibits the Aβ oligomerization and fibril-formation process without showing considerable cytotoxicity in an in vitro neuronal cell culture system. However, in a rat model, we did not detect this phthalocyanine in the brain parenchyma after an intravenous injection (unpublished data), indicating its poor blood–brain barrier (BBB) crossing ability. As the hydrophobicity and overall charge of a molecule play an important role in BBB crossing [26], we modified ZnPc by adding N-methyl-pyridinium groups at eight peripheral β positions to enhance its surface activity and made the molecule cationic to minimize aggregation in an aqueous solution. Previous studies have shown that the interactions of phthalocyanines with proteins or peptides depend on their peripheral structure and central metal ions. For instance, our study revealed that anionic ZnPc containing eight COONa groups interacted differently with the Aβ peptide than that containing 16 COONa [25]. Therefore, altering the composition of the ZnPc periphery could influence its interaction with the Aβ peptide. In this study, we explored the interaction of N-methyl-pyridinium-containing cationic ZnPc (cZnPc) with the Aβ peptide during its fibril-formation process and found that cZnPc effectively inhibited Aβ fibril formation in an in vitro fibril-formation condition.

## 2. Results

### 2.1. Effects of cZnPc on the Aβ_1-42_ Fibril-Formation Process

To investigate the effects of cZnPc on the Aβ_1-42_ fibril-formation process, we incubated 15 µM of the peptide in a fibril-formation buffer alone or with increasing concentrations of cZnPc for 24 h. The fibril formed after the incubation was evaluated using a ThT fluorescence assay. The results demonstrated that after incubating Aβ_1-42_ alone in a fibril-formation buffer for 24 h, a considerable amount of fibril was formed. cZnPc dose-dependently inhibited such Aβ_1-42_ fibril formation, with a stable inhibitory effect observed from a concentration as low as 1 µM (Figure 2A).

Next, we evaluated the effects of cZnPc on Aβ_1-42_ fibril-formation kinetics. Aβ_1-42_ (15 µM) was incubated in the absence or presence of 2 µM cZnPc in a fibril-formation buffer. The incubation was continued from 0 to 48 h, and fibril formation was assessed using a ThT fluorescence assay. Both Aβ_1-42_-alone and Aβ_1-42_ + cZnPc conditions exhibited sigmoidal kinetics, but the rate of fibril formation was different. Aβ_1-42_ alone showed detectable fibrils after 4 h of incubation, reaching a plateau at 16 h. In contrast, when incubated with cZnPc, fibrils were detectable after 8 h, and the plateau was reached at 24 h (Figure 2B). Moreover, the levels of fibrils at the plateau were decreased in Aβ_1-42_ + cZnPc conditions compared to that in Aβ_1-42_ alone.

Subsequently, the effects of cZnPc on the morphology of Aβ_1-42_ fibrils were examined using electron microscopy. When incubated alone for 24 h, Aβ_1-42_ (50 µM) aggregated to form unbranched elongated fibrils with a diameter of approximately 12 to 14 nm (Figure 2C). However, in the presence of cZnPc (4 µM), mature fibrils were rarely seen. Instead, small dot-like aggregates of the peptide with a diameter of about 5 to 6 nm were formed (Figure 2C).

To investigate the effects of cZnPc on the elongation phase of Aβ fibril-formation kinetics, we used pre-formed Aβ_1-42_ fibril seeds to eliminate the lag phase. In both Aβ_1-42_-alone and Aβ_1-42_ + cZnPc conditions, the fibril formed almost immediately without any lag phase with a time-dependent linear increase (r^2^ = 0.93) of the levels. However, the slope of the linear increase was diminished in the Aβ_1-42_ + cZnPc condition (slope = 0.24) compared to that in Aβ_1-42_ alone (slope = 0.68) (Figure 2D).

### 2.2. Effects of cZnPc on Aβ_1-42_ Oligomer Formation

During Aβ fibril formation, the peptide initially aggregates into oligomers during the lag phase, which further aggregate during the elongation phase to make mature fibrils [27]. To investigate the effects of cZnPc on the Aβ_1-42_ fibril-formation process, we examined the oligomerization of the peptide in the presence of cZnPc. Western blotting results showed that when Aβ_1-42_ was incubated alone, a significant amount of oligomers formed after 1 h of incubation, peaked at 2 h, and started decreasing after 4 h (Figure 3A,B). In the presence of cZnPc, a substantial amount of Aβ_1-42_ oligomers was formed after 1 h of incubation, reaching a plateau at 2 h and remaining relatively stable for up to 8 h. Further analysis showed that compared to the Aβ_1-42_-alone condition, cZnPc significantly increased the levels of Aβ_1-42_ oligomers at 4 and 8 h (Figure 3A,B). Dot blot immunoassay results also demonstrated significantly increased levels of oligomers in the Aβ_1-42_ + cZnPc condition compared to those in Aβ_1-42_ alone from 0.5 h to 8 h of incubation (Figure 3C,D).

### 2.3. Binding of cZnPc with Aβ_1-42_ Peptide

To understand the interaction between cZnPc and Aβ_1-42_ peptide during the fibril-formation process, we investigated their binding. After incubation with cZnPc, Aβ_1-42_ was immunoprecipitated using an anti-Aβ antibody, and cZnPc in the immunoprecipitate was detected using a near-infrared scanning system. Near-infrared scanning demonstrated the presence of cZnPc in the immunoprecipitates when anti-Aβ IgG was used, but not in the immunoprecipitates when normal IgG was used. cZnPc signals were detectable from 6.25 µM (Figure 4A).

A molecular docking simulation was also performed to evaluate the binding of cZnPc with Aβ_1-42_, where Aβ_1-42_ monomer, 10-mer, 30-mer, and 50-mer were used as proteins and cZnPc was the substrate. Aβ_1-42_ monomer and 10-mer showed mostly an α-helix conformation, 30-mer as amorphous aggregates, and 50-mer as β-sheet secondary structures (Appendix A). The simulation showed that several amino acids of Aβ_1-42_ species could interact with cZnPc by hydrogen bonding (Figure 4B,C, Appendix A). When the interactions of cZnPc with the species were compared, stronger interactions were observed with 30-mer species where it interacted with 14 amino acids, of which 12 had hydrophobic side chains (Figure 4B,C, Appendix A). Conversely, only four amino acids of the monomer and five amino acids of the 50-mer interacted with cZnPc, of which three had hydrophobic side chains (Figure 4B,C, Appendix A).

### 2.4. Effects of cZnPc on Hydrophobicity during Aβ_1-42_ Fibril Formation

To investigate the effects of cZnPc on the hydrophobic environment during Aβ_1-42_ fibril formation, an ANS fluorescence assay was performed. While both conditions (Aβ_1-42_-alone and Aβ_1-42_ + cZnPc) showed a time-dependent increase in ANS fluorescence, a small but significant decrease in fluorescence was observed in the Aβ_1-42_ + cZnPc condition after 4 h of incubation compared to that in Aβ_1-42_ alone (Figure 5A). Moreover, a blue shift of the emission maxima was observed after incubating Aβ_1-42_ alone for 4 h, which was continued until 24 h. When Aβ_1-42_ was incubated with cZnPc, a similar pattern of a blue shift of the emission maxima was observed (Figure 5B).

### 2.5. Effects of cZnPc on Tyrosine and Intrinsic Fluorescence during the Aβ_1-42_ Fibril-Formation Process

Tyrosine and intrinsic fluorescence spectroscopy are important ways to study protein folding, aggregation, and amyloid formation [28,29]. Hence, time-dependent changes in tyrosine fluorescence were evaluated in the Aβ_1-42_-alone and Aβ_1-42_ + cZnPc conditions. Tyrosine fluorescence was increased in Aβ_1-42_ + cZnPc compared to that in the Aβ_1-42_ alone condition at 0, 8, and 24 h (Figure 6A). To investigate the extent of tyrosine exposure to the solvent, acrylamide quenching experiments were done. The Stern–Volmer plots demonstrated that acrylamide’s concentration-dependent quenching of solvent-exposed tyrosine fluorescence was linear both in Aβ_1-42_-alone and Aβ_1-42_ + cZnPc conditions (Figure 6B). The analysis of Stern–Volmer constants revealed that it was increased in the Aβ_1-42_ + cZnPc condition, at least at a 10 µM concentration of acrylamide (Figure 6C).

An analysis of intrinsic fluorescence revealed that the excitation at 295 nm caused an emission maximum at 590 nm in both the Aβ_1-42_-alone and Aβ_1-42_ + cZnPc conditions (Figure 6D). The emission maximum started to increase after 4 h of incubation in the Aβ_1-42_-alone conditions, which was decreased at 24 h (Figure 6E). In the case of Aβ_1-42_ + cZnPc, the emission maximum started to increase after 24 h of incubation. The fluorescence intensities were significantly higher in the Aβ_1-42_ + cZnPc condition at earlier time points (0 and 1 h) but decreased at 4 and 8 h. At 24 h, the intensity was similar to that in the Aβ_1-42_-alone condition (Figure 6D,E).

### 2.6. Effects of cZnPc on Aβ_1-42_-Induced Neuronal Toxicity in Culture

Finally, the effects of cZnPc on Aβ_1-42_-induced neurotoxicity were examined in a neuronal cell line culture (A1). cZnPc did not show toxicity in the culture, as revealed by cell culture microscopy and MTT cell viability assay (Figure 7A,C). Cell culture microscopy also showed that cZnPc did not affect the morphology of A1 cells in the culture (Figure 7A). However, when A1 cells were cultured with Aβ_1-42_, a significant decrease in viable cell number was observed. Remarkably, the addition of cZnPc along with Aβ_1-42_, starting from a 5 µM concentration, resulted in a significant increase in viable cell number compared to that in the Aβ_1-42_-alone condition (Figure 7B,D).

The hydrogen bonding of cZnPc with the amino acids of Aβ_1-42_ peptide are shown here.

## 3. Discussion

In this study, we investigated the effects of cationic methylated Zn-phthalocyanine (cZnPc) on Alzheimer’s Aβ peptide aggregation and neuronal toxicity. The conversion of the Aβ peptide monomer to its aggregated species increases its toxic properties and is considered the main cause of neurodegeneration in AD [20]. Consequently, molecules that inhibit the aggregation process are considered potential candidates for AD therapy [30,31]. However, these aggregation inhibitors often suffer from poor bioavailability [32], as the brain is one of the least accessible organs for drug delivery due to the presence of the blood–brain barrier (BBB) [32]. The ability of a compound to cross the BBB is linked to its lipophilicity [26]. Increasing hydrophobicity by adding methyl groups could enhance its bioavailability in the brain. With these considerations in mind, we synthesized cationic ZnPc carrying methyl groups at its eight peripheral β positions, which could potentially improve its ability to cross the BBB. Since this cZnPc showed the ability to inhibit the production of mature fibrils and ameliorate Aβ_1-42_-induced neurotoxicity, it holds promise as a potential therapy for AD.

One of the key findings of this study is that cZnPc inhibited mature fibril formation but increased the oligomerization of the peptide, as revealed by a ThT fluorescence assay, a dot blot assay, and Western blotting. The oligomeric form of the peptide is considered toxic and is mainly responsible for neurodegeneration [20]. However, a cell viability assay showed that cZnPc had a protective effect on Aβ_1-42_-induced neurotoxicity. Various mechanisms of Aβ-induced neurotoxicity have been reported, such as Aβ oligomers binding to cell surface receptors and activation of intracellular signaling, or insertion into the cell membrane to form ion channels, leading to subsequent cell death [33]. Specific structural conformations of Aβ oligomers are necessary to perform such receptor- or ion-channel-mediated effects. Indeed, the size of the oligomers is shown to be related to their neurotoxic properties [34]. In our study, electron microscopy results demonstrated that the size of the oligomers differed when the peptide monomer was incubated with cZnPc. Therefore, the oligomers produced in the presence of cZnPc might have a different structural conformation, and interact differently with the cell membrane, resulting in reduced toxicity to neurons.

The aggregation of Aβ under in vitro conditions can be modulated by changing the local environment, such as pH, temperature, ionic strength, and the presence of detergents [35,36,37,38]. Since cZnPc is a cationic molecule, possibly it can alter the microenvironment, ultimately affecting the structural conformation and fibril-formation process. However, the addition of cZnPc did not expose hydrophobic amino acids or change the hydrophobic environment, as evidenced by the ANS fluorescence assay. These results suggest that cZnPc might not affect the fibril-formation process by changing the local microenvironment and exposing the hydrophobic residues of the peptide [39]. Instead, it likely binds directly to the Aβ_1-42_ peptide, thereby modulating the aggregation process. Molecular docking simulations also suggest that the binding of cZnPc with the peptide might be crucial, as different conformations and molecular species of Aβ_1-42_ interact differently with phthalocyanine. Notably, the docking simulation suggested that cZnPc interacts with a greater number of amino acids by hydrogen bonding when the peptide is in an oligomeric conformation. An increased number of hydrogen bonds indicates a stronger binding. This stronger binding of cZnPc with the peptide oligomers could cause a conformational change, which is not suitable for transforming into fibrillar structures. The results of the elongation assay also support this idea, as cZnPc inhibits the progression of the elongation phase, leading to the formation of non-toxic non-fibrillar aggregates [40]. Importantly, most of the amino acids in oligomeric and fibrillar Aβ_1-42_ that interact with cZnPc are at the C-terminus position and contain hydrophobic side chains. The stretches of hydrophobic amino acids at the C-terminus of Aβ_1-42_ are known to be responsible for its aggregation properties [41]. Hence, interactions between cZnPc and this C-terminus stretch of hydrophobic amino acids, particularly in the oligomeric form, could impede the formation of larger oligomers and fibrils, which are known to contribute to neurotoxicity.

To further investigate the involvement of cZnPc in changing the Aβ_1-42_ conformation, an intrinsic tyrosine fluorescence assay was conducted. Aβ peptide contains only one tyrosine residue in its amino acid sequence, allowing us to assess its conformation by evaluating the fluorescence in a solution during different aggregation states. The tyrosine fluorescence of the peptide was demonstrated to be increased by cZnPc in almost all time points, suggesting a conformational state where tyrosine is more accessible. This point is also supported by the increased quenching constant, which can only be achieved by increased accessibility of the water-soluble quencher acrylamide [28]. Studies have shown that proteins devoid of aromatic amino acids and containing high levels of β-sheet structures can emit fluorescence at visible wavelengths when excited with UV-C (<280 nm) and UV-B (280–320 nm) light [42]. As a reason of such phenomena, it has been speculated that peptide electrons delocalized through intramolecular or intermolecular hydrogen bond formation display these long-wavelength electronic transitions [42]. Aβ exhibits such intrinsic fluorescence in the visible range during β-sheet conformational changes [29]. We observed such intrinsic fluorescence when aggregated Aβ was excited at a 295 nm wavelength. Importantly, we found that adding cZnPc initially increased the internal fluorescence at the visible range, but it decreased compared to that in the Aβ-alone condition at a later time when fibrillary Aβ started to appear, suggesting a reduction of β-sheet structure. However, to confirm the change in β-sheet structure of Aβ by cZnPc, circular dichroism spectroscopy would provide more detail information [25]. Nevertheless, this tyrosine fluorescence and internal fluorescence results imply that the aggregated peptide conformation could be different when cZnPc was added. The strong interaction of cZnPc with Aβ_1-42_ oligomers could change the conformation of the aggregated peptide and make it less toxic. Such aggregated Aβ can be cleared by the microglia to prevent its deposition in the brain [43]. Investigating the detailed interaction of the cZnPc-Aβ oligomeric complex with the microglia in vitro and in vivo AD animal models could shed light on the clearance of the peptide.

## 4. Materials and Methods

### 4.1. Materials

Synthetic Aβ_1-42_ was obtained from the Peptide Institute (Osaka, Japan). The peptide was dissolved in 0.1% ammonia water at a concentration of 250 µM on ice, immediately aliquoted, and stored at −70 °C for further use. After dissolution in the solvent, chromatographic data indicated a monomeric form of the peptides, as stated by the manufacturer. Dulbecco’s modified Eagle’s medium (DMEM, high glucose) and filter-sterilized deionized water were purchased from Wako Pure Chemicals (Richmond, VA, USA). Fetal bovine serum (FBS), L-glutamine, and antibiotics–antimycotics for the cell culture were obtained from Invitrogen (Carlsbad, CA, USA). Thioflavin T (ThT) was obtained from Sigma-Aldrich (St Louis, MO, USA). The nitrocellulose membrane for the dot blot assay was from Millipore (Billerica, MA, USA).

### 4.2. Synthesis of N-Methyl-Pyridinium Containing Cationic ZnPc (cZnPc)

The detailed synthesis of 2,3,6,7,10,11,14,15-Octakis-[(4-methyl-3-pyridyloxy) phthalocyaninato] zinc(II) (2Zn) or cZnPc was previously described in our report [44]. In summary, a macrocyclization reaction was carried out using phthalonitrile 3 (0.45 g, 1.3 mmol) and zinc acetate dihydrate (0.14 g, 0.64 mmol). The resulting cZnPc was obtained as a blue-green solid powder. Chemical and spectrometric analysis data, as well as its crystal structure (CCDC number: 1869572), are documented in the same report [44].

### 4.3. Aβ Peptide Fibril Formation

For fibril formation, solutions containing indicated concentrations of synthetic Aβ_1-42_ peptide and fibril-formation buffer (50 mM phosphate buffer pH 7.5 and 100 mM NaCl) with or without cZnPc at the specified concentrations were prepared. The reaction mixtures were incubated at 37 °C for the indicated times without agitation. After incubation, the reaction was terminated by quickly freezing the samples and stored at −70 °C until fibril measurement. Since phthalocyanines are photoreactive and produce singlet oxygen, all experiments were done in dark conditions.

### 4.4. Assessment of Aβ Fibril by ThT Fluorescence Assay

The presence of the fibril structure in the samples was evaluated using ThT fluorescence spectroscopy. To measure fibril levels, the samples were diluted tenfold with glycine (pH 8.5, 50 mM final concentration) and ThT (5 µM final concentration), and fluorescence was measured using a fluorescence spectrophotometer (F2500 spectrofluorimeter, Hitachi, Tokyo, Japan). The excitation and emission wavelengths were set to 446 nm and 490 nm, respectively. To obtain a normalized value, the fluorescence intensity of the buffer containing the same concentration of cZnPc was subtracted from that of the sample.

### 4.5. Electron Microscopy

Electron microscopy was performed as previously described [25]. Briefly, Aβ_1-42_ (50 µM) with or without cZnPc (4 µM) was incubated in a fibril-formation buffer for 24 h to prepare fibrils. After fibril formation, approximately 5 µL of a sample was applied to a carbon-coated Formvar grid (Nisshin EM, Tokyo, Japan) and incubated for 1 min. Then, an equal volume of a 0.5% *v*/*v* glutaraldehyde solution was added to the sample droplet and incubated for 1 additional minute. After incubation, excess fluid was soaked off. Subsequently, 10 µL of a 2% *w*/*v* uranyl acetate solution was added to the grid and incubated for 2 min. The grid was then placed on a water drop and incubated for 1 min to wash the excess uranyl acetate. The grid was taken out from the water, air-dried, and examined under an electron microscope (EM-002B, Topcon, Tokyo, Japan).

### 4.6. Tyrosine Fluorescence Assay

The tyrosine fluorescence assay was performed as previously described [28]. Briefly, 25 µM of Aβ_1-42_ fibril was prepared with 0 or 2 µM cZnPc for the indicated times. The samples were diluted 10 times with PBS, and fluorescence intensity was measured using a Hitachi F2500 spectrofluorimeter with an excitation wavelength of 275 nm and emissions scanned from 290 to 360 nm at 60 nm/min. The excitation and emission slit widths were 5 and 10 nm, respectively. The emission spectra of the samples were subtracted from that of the buffer spectra to obtain a normalized value.

To assess tyrosine fluorescence quenching, increasing concentrations of acrylamide were added to the samples immediately before measurement. The emission spectra were normalized with the spectra of the buffer containing the same concentrations of acrylamide. The data were analyzed using the Stern–Volmer equation: F0/F = 1 + Ksv[Q], where F0 and F are the fluorescence intensities in the absence and presence of acrylamide, [Q] is the concentration of acrylamide, and Ksv is the Stern–Volmer constant.

### 4.7. Internal Fluorescence Spectroscopy

The internal fluorescence of the Aβ_1-42_ peptide was measured using a Hitachi F2500 spectrofluorimeter. Aβ_1-42_ (15 µM) peptide in the absence or presence of cZnPc was incubated in a fibril-formation buffer for the indicated times. The samples were then diluted with MilliQ water to reduce the peptide concentration to 1.5 µM. The excitation wavelength was set at 295 nm, and fluorescence emission was scanned in the range of 380–700 nm. Slit widths for excitation and emission were 5 nm. The fluorescence emission spectrum of buffer only or cZnPc-containing buffer (background intensity) was subtracted from the emission spectrum of the samples. The emission maximum data are presented as an arbitrary fluorescence unit.

### 4.8. ANS Fluorescence Spectroscopy

The fluorescence intensity changes of 8-anilino-1-naphthalene sulfonic acid (ANS) were used to evaluate the relative exposure levels of hydrophobic surfaces of Aβ_1-42_ in fibril-formation conditions [19]. Aβ_1-42_ (25 µM) was incubated with 0 or 2 µM cZnPc for the indicated times in a fibril-formation condition. After incubation, ANS (5 µM) was added to the samples, and fluorescence intensity measurements were obtained using a Hitachi F2500 spectrofluorimeter, where the excitation wavelength was set at 360 nm, and the emission spectra were scanned from 380 to 600 nm. Slit widths for excitation and emission were 5 nm. The intensity and the wavelength of the emission maximum are presented as the mean of three independent experiments.

### 4.9. Elongation Assay

The elongation assay was performed as described previously [19]. In brief, a 50 µM concentration of the Aβ_1-42_ monomer was incubated in a fibril-formation buffer at 37 °C for 48 h to prepare fibrils. The entire reaction mixture was then sonicated for 10 min to make protofibrils. To eliminate the lag phase and analyze the elongation phase, 15 µM (3 µg) of synthetic Aβ_1-42_ monomer was incubated in a fibril-formation buffer in the presence of 0.2 µg of sonicated Aβ_1-42_ fibrils in a total volume of 50 µL at 37 °C for the indicated times. The fibril formed after the incubation was analyzed by measuring the ThT fluorescence of the samples. ThT fluorescence of buffer containing 0.2 µg of sonicated Aβ_1-42_ fibrils was subtracted from the fluorescence of the samples

### 4.10. Aβ Dot Blot Assay for Oligomer Analysis

For Aβ_1-42_ oligomer detection, a dot blot immunoassay was performed using an oligomer-specific antibody [25]. First, 50 µM of Aβ_1-42_ was incubated in the absence or presence of cZnPc (5 µM) for the indicated times. Then, 2 μg equivalent of Aβ_1-42_ peptide samples were spotted on a nitrocellulose membrane. The membrane was blotted with an oligomer-specific antibody (A11, Invitrogen). To detect the immunoreactive peptide, an infrared dye-conjugated species-specific secondary antibody was used, and the signals were detected using an Odyssey infrared dye scanning system (Li-Cor Biosciences, Lincoln, NE, USA).

### 4.11. Aβ Western Blotting for Oligomer Analysis

Aβ Western blotting for oligomer analysis was done as described previously [45]. Briefly, Aβ_1-42_ monomers (50 µM) were incubated in the absence or presence of cZnPc (5 µM) in a fibril-formation buffer for the indicated times. After incubation, the 2 µg equivalent of the peptide was separated by 4–20% gradient SDS–polyacrylamide gel electrophoresis in non-reducing and non-denaturing conditions and transferred to PVDF membranes (Millipore). Then, the membrane was immunoblotted with an anti-Aβ antibody (mouse monoclonal, clone B4, Santa Cruz). Immunoreactive protein was detected using infrared dye-conjugated anti-mouse IgG and the Odyssey infrared dye scanning system (Li-Cor) according to the manufacturer’s instructions. Densitometric analysis of the oligomers excluding dimers and trimers was done using ImageJ (ij154-win-java8).

### 4.12. Aβ_1-42_ and cZnPc Docking Simulations

To assess the interaction of the Aβ_1-42_ peptide and cZnPc, a molecular docking simulation was performed using Molegro Virtual Docker (Molexus IVS, Odder, Denmark). The PDB files of Aβ_1-42_ 10-mer (1IYT), 30-mer (1Z0Q), and 50-mer (2BEG) were downloaded from the RCSB protein data bank. A PDB file of the Aβ_1-42_ monomer was generated by splitting a monomer from a 10-mer (1IYT) using Molegro Virtual Docker. The crystal structure of cZnPc (CCDC number: 1869572) was used as a ligand. The interactions of cZnPc and molecular species of Aβ_1-42_ were evaluated by analyzing MolDock scores, reRank scores, and hydrogen bonding with amino acids of the species.

### 4.13. Cell Culture

A human neuronal cell line (A1) was used to evaluate the toxicity of cZnPc and the Aβ_1-42_ peptide. The A1 cell line was generated by somatic fusion between a human fetal cerebral neuron and a human neuroblastoma cell, as previously described [46]. The cell line exhibits characteristic morphological, electrophysiological, and expressional features of neurons in culture. A1 cells were cultured in DMEM containing 5% FBS, L-glutamine, and antibiotics–antimycotics. During stimulation with Aβ_1-42_ and cZnPc, the concentration of FBS was reduced to 1%. Photomicrographs of the cultured cells were obtained using an inverted cell culture microscope (Olympus CK-2, Olympus, Tokyo, Japan) equipped with a digital photography system (Olympus).

### 4.14. MTT Cell Viability Assay

The effect of cZnPc on A1 cell viability and Aβ_1-42_-induced toxicity was evaluated by an MTT cell viability assay, as described previously [25]. Briefly, A1 cells (5 × 10^3^/well) were seeded into the wells of a 96-well cell culture plate and cultured for 24 h. The cells were then treated with the indicated concentrations of cZnPc, Aβ_1-42_, or both Aβ_1-42_ and cZnPc in 100 µL of DMEM containing 1% FBS for 48 h. After appropriate treatment, 20 µL of MTT solution (Sigma-Aldrich) (5 mg/mL) was added to the culture and further incubated for 3.5 h. Then, the medium was completely removed carefully without disturbing the cells, and MTT solvent (4 mM HCl, 0.1% Nonidet P-40 in isopropanol) was added. The plate was incubated for 15 min at room temperature with protection from light. Then, the absorbance was read at 590 nm using an ELISA plate reader. The absorbance of the cells under normal culture conditions was used as a control, and the results are expressed as percentage of control.

### 4.15. Statistical Analysis

The numerical data are expressed as the mean ± SD of at least three independent experiments. Statistical analysis for comparing mean values was performed using one-way ANOVA, followed by Scheffe’s post hoc test. Linear regression was done using Microsoft Excel (Office 365). The fibril-formation kinetics were analyzed using SigmaPlot software (SigmaPlot 11, Systat Software Inc., San Jose, CA, USA). *p* values <0.05 indicated statistical significance.

## 5. Conclusions

In conclusion, our results suggested that methylated cZnPc has the ability to bind to oligomeric Aβ_1-42_ peptide and prevent subsequent fibril elongation. The cZnPc-Aβ oligomeric complexes have altered structures and decreased neurotoxicity. We believe that cZnPc could be a potential candidate for the treatment of AD.

## Figures and Tables

**Figure 1 ijms-25-08931-f001:**
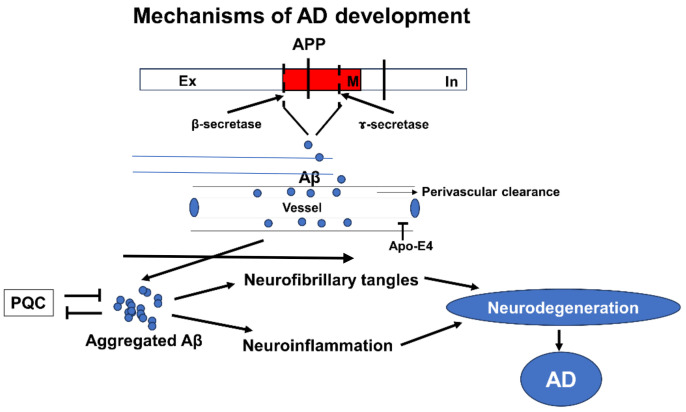
Mechanisms of AD development. Aβ plays a central role in AD development. The peptide is produced from the transmembrane protein APP by the actions of β- and γ-secretases. The Aβ peptide is usually cleared from the brain through cell-mediated or perivascular-mediated clearance. Apo-E plays an important role in perivascular-mediated clearance, with the Apo-E4 isoform hampering Aβ clearance. When excess Aβ accumulates in the brain, either due to excessive production or decreased clearance, it tends to aggregate. Members of the protein quality control (PQC) system, including the ubiquitin–proteasome system, activate to mitigate such changes. However, aggregated Aβ also has an inhibitory effect on PQC. When the balance shifts towards PQC inhibition from its activation, more aggregated Aβ accumulates and aggregates. The aggregated peptide directly causes neurodegeneration. It also induces tau hyperphosphorylation and neurofibrillary tangle formation, along with neuroinflammation. These changes contribute to neurodegeneration, ultimately leading to the formation of AD lesions. APP = amyloid precursor protein; Ex = extracellular; In = intracellular; M = membranous.

**Figure 2 ijms-25-08931-f002:**
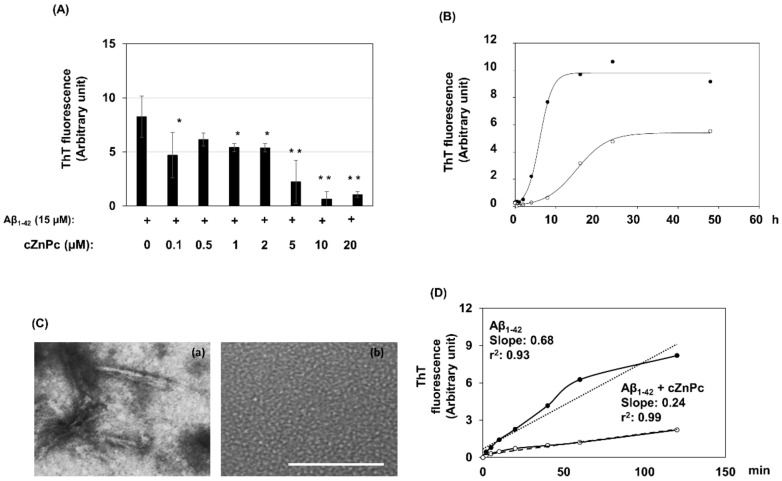
Effects of cZnPc on the fibril-formation process of the Aβ_1-42_ peptide. (**A**) Aβ_1-42_ (15 µM) was incubated with the indicated concentrations of cZnPc in a fibril-forming buffer for 24 h. Fibril formation after incubation was evaluated by a ThT fluorescence assay as described in the Materials and Methods. The fluorescence values (arbitrary unit) are presented as the means ± SD of at least three independent experiments. The symbols indicate statistically significant differences compared to the Aβ_1-42_-alone condition (* *p* < 0.05, ** *p* < 0.01). (**B**) To check the effects on fibril-formation kinetics, Aβ_1-42_ (15 µM) was incubated alone or with cZnPc (2 µM) in a fibril-formation buffer for indicated times. The total fibril formed at the end of incubation was evaluated by a ThT fluorescence assay. (**C**) The morphology of the aggregates after incubation of the Aβ_1-42_ peptide (50 µM) alone (**a**) or with cZnPc (4 µM) (**b**) in a fibril-formation buffer for 24 h was evaluated by transmission electron microscopy. Scale bar = 100 nm. (**D**) To check the effects of cZnPc on the elongation phase of fibril-formation kinetics, an Aβ_1-42_ fibril seed (0.2 µg) was added to 3 µg (15 µM) of the peptide, and the peptide was allowed to form fibrils in the absence or presence of cZnPc (2 µM) for indicated times. After incubation, the fibrils in the samples were measured by a ThT fluorescence assay. The average fluorescence intensities of four independent elongation assay experiments are shown here.

**Figure 3 ijms-25-08931-f003:**
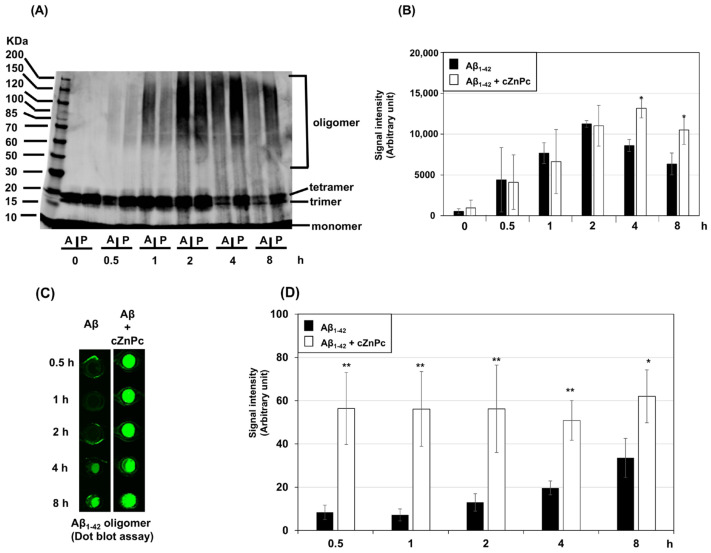
Effects of cZnPc on the oligomerization of Aβ_1-42_ peptide. Aβ_1-42_ peptide (50 µM) was incubated alone or in the presence of cZnPc (4 µM) in a fibril-formation buffer for indicated times, and 2 µg peptide equivalent samples were used to analyze the aggregation states by Western blotting, as described in the Materials and Methods. Representative Western blotting data are shown in (**A**), where “A” indicates Aβ_1-42_ alone, and “P” indicates the Aβ_1-42_ + cZnPc condition. Aggregated peptide oligomers, except low-molecular-weight species, were quantified by densitometry, and the average of 4 individual experiments is shown in (**B**). Oligomerization was further evaluated by a dot blot immunoassay, where 2 µg equivalent Aβ_1-42_ peptide alone or in the presence of cZnPc (4 µM) was incubated in fibril-formation conditions for indicated times and spotted on a nitrocellulose membrane. The membrane was immunoblotted with an oligomer-specific antibody. Representative dot blot data are shown in (**C**). The dot blot data were quantified by densitometry, and the averages of 4 individual experiments are shown in (**D**). Statistical significance is denoted as follows: * *p* < 0.05 vs. Aβ_1-42_-alone condition at the same time point; ** *p* < 0.01 vs. Aβ_1-42_-alone condition at the same time point.

**Figure 4 ijms-25-08931-f004:**
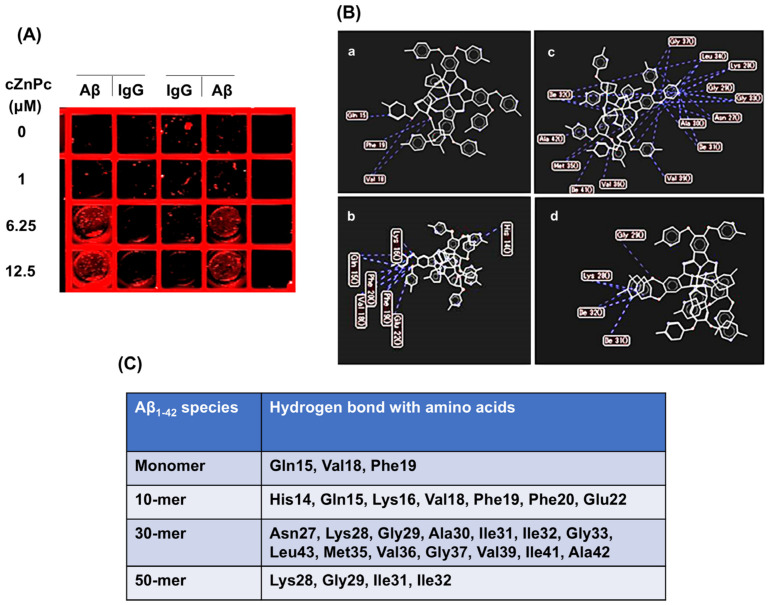
Binding of cZnPc with the Aβ_1-42_ peptide. (**A**) To evaluate the binding ability of cZnPc with Aβ_1-42_, 50 µM of the peptide was incubated with the indicated concentration of cZnPc for 24 h in a fibril-formation buffer. After incubation, 10 µg of peptide equivalent was diluted 10 times with PBS, and Aβ_1-42_ was immunoprecipitated with an anti-Aβ antibody, as described in the Materials and Methods. As a control, Aβ_1-42_ was immunoprecipitated with normal mouse IgG instead of an anti-Aβ antibody. The immunoprecipitates were taken in an ELISA plate, and the presence of cZnPc was evaluated by near-infrared scanning at 680 nm. A scanning picture is shown here. (**B**) To further evaluate the interaction, a docking simulation was performed using Molegro Virtual Docker, where the structures of various Aβ_1-42_ species were used as a protein and that of cZnPc as a ligand. The hydrogen bonding of cZnPc with the amino acids of Aβ_1-42_ peptide are shown here. (**C**) The list of the amino acids of Aβ_1-42_ species that interacted with cZnPc by hydrogen bonding are given here.

**Figure 5 ijms-25-08931-f005:**
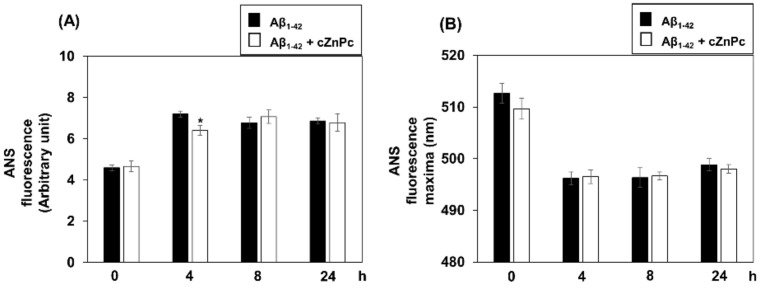
Effects of cZnPc on the hydrophobicity of the Aβ_1-42_ peptide during fibril formation. To check the effects of cZnPc on the hydrophobic microenvironment, Aβ_1-42_ (25 µM) was incubated in the absence or presence of cZnPc (2 µM) in a fibril-forming buffer for indicated times, and ANS fluorescence spectroscopy was done as described in the Materials and Methods. Time-dependent changes in fluorescence intensities are shown in (**A**), and the wavelengths of the maximum are shown in (**B**). Statistical significance is denoted as follows: * *p* < 0.05 vs. corresponding Aβ_1-42_-alone condition.

**Figure 6 ijms-25-08931-f006:**
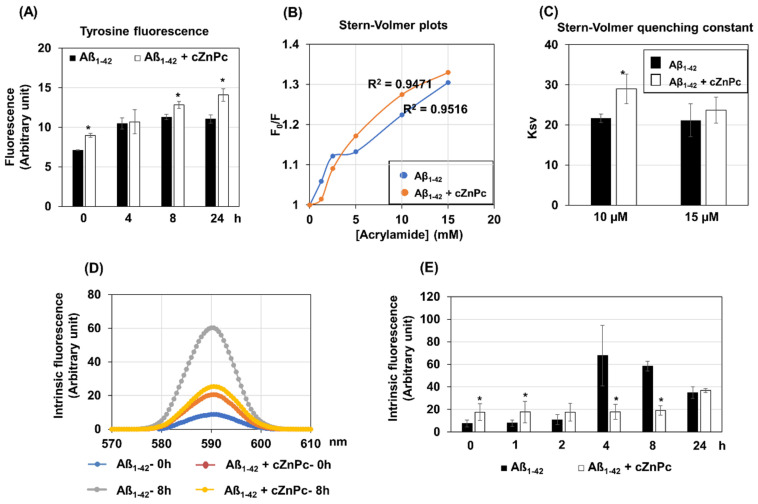
Effects of cZnPc on tyrosine and intrinsic fluorescence of the Aβ_1-42_ peptide during fibril formation. To analyze the structural changes, Aβ_1-42_ (25 µM) was allowed to make fibrils in the absence or presence of cZnPc (2 µM) for indicated times, and the tyrosine fluorescence of the samples was measured. (**A**) shows the time-dependent changes of tyrosine fluorescence of Aβ_1-42_ alone and Aβ_1-42_ + cZnPc. In (**B**), a Stern–Volmer plot of acrylamide quenching of tyrosine fluorescence is shown, where Aβ_1-42_ alone or Aβ_1-42_ + cZnPc were incubated for 24 h. The quenching constants at 10 and 15 µM concentrations of acrylamide are shown in (**C**). (**D**) To further analyze the structural changes, the internal fluorescence of the samples was measured. Aβ_1-42_ (15 µM) was allowed to make fibrils in the absence or presence of cZnPc for indicated times, and the internal fluorescence of the samples was measured. Representative buffer-normalized fluorescence spectra are shown here. (**E**) The averages of intrinsic fluorescence intensities are shown here. The numerical data are presented here as an average ± SD of at least 4 independent experiments. Statistical significance is denoted as follows: * *p* < 0.05 vs. corresponding Aβ_1-42_-alone condition.

**Figure 7 ijms-25-08931-f007:**
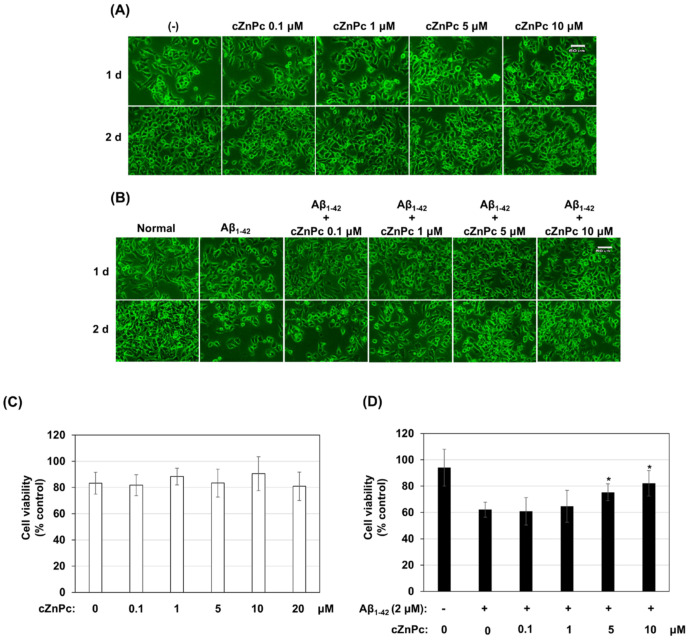
Effects of cZnPc on Aβ_1-42_-induced neuronal toxicity in a culture. Cells of a neuronal cell line (A1) were cultured in a 1% FBS/DMEM medium containing indicated concentrations of cZnPc for 2 days. The cells were monitored by microscopy, and the photomicrographs of the cultured cells are shown in (**A**). To analyze the viability, an MTT assay was done, and the average results are shown in (**C**). (**B**) A1 cells were cultured in a 1% FBS/DMEM medium containing 2 µM of Aβ_1-42_ and indicated concentrations of cZnPc for 2 days. The cells were monitored by microscopy, and the photomicrographs of the cultured cells are shown here (**B**). To further analyze the effects of cZnPc on Aβ1-42-induced toxicity, an MTT assay was done, and the average results are shown in (**D**). The numerical data are presented here as an average ± SD of at least 4 independent experiments. Statistical significance is denoted as follows: * *p* < 0.05 vs. Aβ_1-42_ alone condition. Scale bar = 60 µm.

## Data Availability

All the data of this study are included in the manuscript or in the Appendix A.

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
