# Peer review of "A Cationic Zn-Phthalocyanine Turns Alzheimer’s Amyloid β Aggregates into Non-Toxic Oligomers and Inhibits Neurotoxicity in Culture"

_ijms, 2024, doi:10.3390/ijms25168931_

Round 1

Reviewer 1 Report

Comments and Suggestions for Authors

The article by Abdullah Md. Sheikh and colleagues provide a detailed overview of cationic Zn-phthalocyanine causing Alzheimer’s amyloid β aggregates to non-toxic oligomers and inhibit neurotoxicity in culture. In this study, authors have explored the interaction of a cationic methylated Zn-phthalocyanine (cZnPc) with the Aβ peptide based on their previous study findings. So, the main question addressed by this research is whether N-methylpyridinium-containing cationic ZnPc (cZnPc) inhibits Aβ fibril formation?

The original part of this study was defining the role of methylated cZnPc, which binds to oligomeric Aβ1-42 peptide and prevents subsequent fibril elongation. The authors have used multiple techniques such as ThT Fluorescence Assay, Electron Microscopy, Tyrosine Fluorescence Assay, Internal Fluorescence Spectroscopy, ANS Fluorescence Spectroscopy, Elongation Assay, Dot Blot Assay, Western Blotting, Docking Simulations, MTT Assay etc., to study different effects on Aβ peptide and its role in Alzheimer's disease (AD).

Overall, this is a well-written article and provides sufficient experimental results to conclude the possible role of methylated cZnPc to bind to oligomeric Aβ1-42 peptide and prevent subsequent fibril elongation, which can be used as a potential candidate for AD treatment. However, the authors will need to address the following points to strengthen the manuscript further.

1.In the introduction section, the authors need to provide a more detailed background and should include other protein's roles in AD and then focus more on the Aβ peptide. It would be beneficial if a figure is incorporated in the introduction section.

2.The authors have performed different sets of experiments to come to a possible conclusion. However, Circular dichroism (CD) spectroscopy techniques are needed, which can be further used to determine the structure and folding of Aβ. The results obtained from these experiments will help to confirm the findings in a more confident way.

3.The conclusions described are almost consistent with the evidence and arguments presented. For example, the presence of fibril structure was confirmed by ThT fluorescence spectroscopy. Other techniques, such as ANS (8-anilino-1-naphthalene sulfonic acid) fluorescence assay, were used to evaluate the relative exposure levels of hydrophobic surfaces of Aβ1-42 in fibril formation. A similar easy assay, such as the Elongation assay dot blot assay, helps to find the fibril formation and western bloating to find the aggregation state. However, more experiments are required using living-cell models, preferably neuronal cells, and authors need to validate their findings in this system. This experiment will be a crucial part of confirming the study's findings.

4.In Figure 1A, the font size should be increased.
In Figure 1C, the scale bar is missing.
In Figure 2A, the marker lanes molecular weight is missing. Figure 3B, a high-resolution file is needed.
Overall, in most of the figures, the authors should be consistent with naming the X/Y axis e.g., h. Also, significance should be mentioned throughout the manuscripts whenever statistical analysis was done, e.g., ns, *, **, ***, etc.
It would be a good practice if the authors used the same color consistency throughout the manuscript for Aβ1-42, Aβ1-42 + cZnPc.
The scale bar is not visible in Figure 6A-B.
A full stop should be removed from the title.

5.The reference list is comprehensive and includes relevant publications. The authors must include the most recent publications in this article and discuss the findings with the current study's results.

Author Response

Reviewer 1

The article by Abdullah Md. Sheikh and colleagues provide a detailed overview of cationic Zn-phthalocyanine causing Alzheimer’s amyloid β aggregates to non-toxic oligomers and inhibit neurotoxicity in culture. In this study, authors have explored the interaction of a cationic methylated Zn-phthalocyanine (cZnPc) with the Aβ peptide based on their previous study findings. So, the main question addressed by this research is whether N-methylpyridinium-containing cationic ZnPc (cZnPc) inhibits Aβ fibril formation?

The original part of this study was defining the role of methylated cZnPc, which binds to oligomeric Aβ1-42 peptide and prevents subsequent fibril elongation. The authors have used multiple techniques such as ThT Fluorescence Assay, Electron Microscopy, Tyrosine Fluorescence Assay, Internal Fluorescence Spectroscopy, ANS Fluorescence Spectroscopy, Elongation Assay, Dot Blot Assay, Western Blotting, Docking Simulations, MTT Assay etc., to study different effects on Aβ peptide and its role in Alzheimer's disease (AD).

Overall, this is a well-written article and provides sufficient experimental results to conclude the possible role of methylated cZnPc to bind to oligomeric Aβ1-42 peptide and prevent subsequent fibril elongation, which can be used as a potential candidate for AD treatment. However, the authors will need to address the following points to strengthen the manuscript further.

  1. In the introduction section, the authors need to provide a more detailed background and should include other protein's roles in AD and then focus more on the Aβ peptide. It would be beneficial if a figure is incorporated in the introduction section.

Response: According to the reviewer’s suggestion, we have added the importance of other proteins in AD pathology, and discussed the central role of Aβ in that pathology. We also added a schematic figure that show the importance of the proteins related to AD and their relation with Aβ.

  1. The authors have performed different sets of experiments to come to a possible conclusion. However, Circular dichroism (CD) spectroscopy techniques are needed, which can be further used to determine the structure and folding of Aβ. The results obtained from these experiments will help to confirm the findings in a more confident way.

Response: We also agree with the reviewer that CD spectroscopy is an important technique to analyze the secondary structure of the proteins. However, the CD spectroscopy system is not available at our campus at present. Although not as accurate as CD spectroscopy, changes in tyrosine fluorescence and intrinsic fluorescence (excitation with UV, and emission at visible wavelength) during fibril formation process can also be used to analyze the conformational changes and β-sheet formation. Since we did not used CD spectroscopy for secondary structural changes, we decided to use tyrosine fluorescence and intrinsic fluorescence for analysis of conformational changes and β-sheet structure. However, we think this point the reviewer raised is important. So, in the revised manuscript, we discussed this point.

  1. The conclusions described are almost consistent with the evidence and arguments presented. For example, the presence of fibril structure was confirmed by ThT fluorescence spectroscopy. Other techniques, such as ANS (8-anilino-1-naphthalene sulfonic acid) fluorescence assay, were used to evaluate the relative exposure levels of hydrophobic surfaces of Aβ1-42 in fibril formation. A similar easy assay, such as the Elongation assay dot blot assay, helps to find the fibril formation and western bloating to find the aggregation state. However, more experiments are required using living-cell models, preferably neuronal cells, and authors need to validate their findings in this system. This experiment will be a crucial part of confirming the study's findings.

Response: We also think that the effects of cZnPc on cell culture, and its effects on Aβ-induced cell toxicity is important. Hence, we have used cell culture models to see the toxicity of the cZnPc, and its effects on Aβ-induced neuronal toxicity.

  1. In Figure 1A, the font size should be increased.

Response: According to the reviewer’s comment, the font size of Figure 1A and other figures was increased.

  1. In Figure 1C, the scale bar is missing.

Response: Scale bar is added in figure 1C.

  1. In Figure 2A, the marker lanes molecular weight is missing. Figure 3B, a high-resolution file is needed.

Response: Molecular weights of the markers are added in Figure 2A. Also, figure 3B was replaced with high resolution pictures.

  1. Overall, in most of the figures, the authors should be consistent with naming the X/Y axis e.g., h. Also, significance should be mentioned throughout the manuscripts whenever statistical analysis was done, e.g., ns, *, **, ***, etc.

Response: We have corrected the figures according to the reviewer’s suggestion.

  1. It would be a good practice if the authors used the same color consistency throughout the manuscript for Aβ1-42, Aβ1-42 + cZnPc.

Response: According to the reviewer’s suggestion, we have changed the figures.

  1. The scale bar is not visible in Figure 6A-B.

Response: We have added the scale bar in the revised figure.

  1. A full stop should be removed from the title.

Response: We have corrected the mistake in the revised manuscript.

  1. The reference list is comprehensive and includes relevant publications. The authors must include the most recent publications in this article and discuss the findings with the current study's results.

Response: According to the reviewer’s suggestion, we have checked and corrected the references.

Reviewer 2 Report

Comments and Suggestions for Authors

In the submitted manuscript, Nagai et al. continued their previous work (FEBS Journal 282 (2015) 463–476), synthesizing a cationic form of the previously investigated compound with the aim of obtaining a compound that can pass the blood-brain barrier. They demonstrated that the previous anionic form of the compound can interact with beta-amyloid. In the submitted paper, they investigated its interaction with this peptide in detail using various physical-chemical and biophysical methods.

In addition to several performed experiments, the authors also investigated the effect of their compound, cZnPc, on the hydrophobicity of the environment during fibril formation. They revealed a significant increase in fluorescence intensity combined with a bathochromic shift under both conditions after 4 hours. However, in the presence of cZnPc, the fluorescence intensity was lower compared to bare amyloid (Figure 2A). Although the authors graphically presented how the emission maximum shifted with incubation time in Figure 4B, they did not explain these results in the manuscript. This explanation should be included.

Furthermore, during the investigation of internal fluorescence (probably the authors meant intrinsic fluorescence), the authors used 295 nm as the excitation wavelength and recorded emission spectra between 380-700 nm. Capturing emission spectra in this range after excitation at 295 nm is unusual because, with excitation of Trp residues in proteins (λ_ex 295 nm), the emission spectra are typically in the range of 320-360 nm. I assume that the authors wanted to detect the amyloid aggregation process and attempted to find appropriate physical-chemical methods. If the authors intended to perform a similar experiment as those in the cited manuscript (Analyst, 2013, 138, 2156), I wonder why they used 295 nm as the excitation wavelength instead of 450 or 405 nm. Please explain this.

Minor remarks: Figure 5B – The intercept in the SV plot has to be 1, as stated in the SV equation. Therefore, the plot presented in Figure 5B should be corrected.

Author Response

Reviewer 2

  1. In the submitted manuscript, Nagai et al. continued their previous work (FEBS Journal 282 (2015) 463–476), synthesizing a cationic form of the previously investigated compound with the aim of obtaining a compound that can pass the blood-brain barrier. They demonstrated that the previous anionic form of the compound can interact with beta-amyloid. In the submitted paper, they investigated its interaction with this peptide in detail using various physical-chemical and biophysical methods.

In addition to several performed experiments, the authors also investigated the effect of their compound, cZnPc, on the hydrophobicity of the environment during fibril formation. They revealed a significant increase in fluorescence intensity combined with a bathochromic shift under both conditions after 4 hours. However, in the presence of cZnPc, the fluorescence intensity was lower compared to bare amyloid (Figure 2A). Although the authors graphically presented how the emission maximum shifted with incubation time in Figure 4B, they did not explain these results in the manuscript. This explanation should be included.

Response: We have checked the Figure 2A and 4B, and corrected the description of the results in the revised manuscript.

  1. Furthermore, during the investigation of internal fluorescence (probably the authors meant intrinsic fluorescence), the authors used 295 nm as the excitation wavelength and recorded emission spectra between 380-700 nm. Capturing emission spectra in this range after excitation at 295 nm is unusual because, with excitation of Trp residues in proteins (λ_ex 295 nm), the emission spectra are typically in the range of 320-360 nm. I assume that the authors wanted to detect the amyloid aggregation process and attempted to find appropriate physical-chemical methods. If the authors intended to perform a similar experiment as those in the cited manuscript (Analyst, 2013, 138, 2156), I wonder why they used 295 nm as the excitation wavelength instead of 450 or 405 nm. Please explain this.

Response: First of all, amino acid tryptophan is not present in Aβ peptide. The amino acid sequence of Aβ that we have used in our study is the following, ‘Asp-Ala-Glu-Phe-Arg-His-Asp-Ser-Gly-Tyr-Glu-Val-His-His-Gln-Lys-Leu-Val-Phe-Phe-Ala-Glu-Asp-Val-Gly-Ser-Asn-Lys-Gly-Ala-Ile-Ile-Gly-Leu-Met-Val-Gly-Gly-Val-Val-Ile-Ala’. The sequence was provided by the manufacturer of the peptide. For detection of intrinsic fluorescence, the peptide needs to be excited with lights of UV spectra. If β-sheet is present in the sample, we can see an emission of light at visible wavelength. In the mentioned paper, they first wanted to identify an excitation wavelength. So, they fixed an emission spectrum at visible wavelength, specifically at 470 nm, then they scanned for excitation spectra, which wavelength gives maximum quantum yield. Similarly, they fixed the excitation wavelength at 370 nm to find out which wavelength gives the maximum quantum yield. They have found that in their system the excitation and emission maxima are at 355 nm and 440 nm respectively (Figure 4 of Analyst, 2013, 138, 2156). In our system of fibril formation and measurement, we found that the maximum emission we can find at 295 nm excitation wavelength. So, we used 295 nm wavelength as excitation in our intrinsic fluorescence experiments. Since this point is important, we have discussed it in the revised manuscript.

Minor remarks: Figure 5B – The intercept in the SV plot has to be 1, as stated in the SV equation. Therefore, the plot presented in Figure 5B should be corrected.

Response: according to the reviewer’s suggestion, Figure 5B has been corrected.
